# Comparative Analyses of Bioactive Compounds in *Inonotus obliquus* Conks Growing on *Alnus* and *Betula*

**DOI:** 10.3390/biom12091178

**Published:** 2022-08-25

**Authors:** Rein Drenkhan, Hedi Kaldmäe, Maidu Silm, Kalev Adamson, Uko Bleive, Alar Aluvee, Mart Erik, Ain Raal

**Affiliations:** 1Chair of Silviculture and Forest Ecology, Institute of Forestry and Engineering, Estonian University of Life Sciences, 51006 Tartu, Estonia; 2Polli Horticultural Research Centre, Chair of Horticulture, Institute of Agricultural and Environmental Sciences, Estonian University of Life Sciences, 69108 Polli, Estonia; 3Chair of Hydrobiology and Fisheries, Institute of Agricultural and Environmental Sciences, Estonian University of Life Sciences, 51006 Tartu, Estonia; 4Inopure OÜ, 51010 Tartu, Estonia; 5Faculty of Medicine, Institute of Pharmacy, University of Tartu, 50411 Tartu, Estonia

**Keywords:** betulinic acid, betulin, inotodiol, lanosterol, α- and β-glucan, *Alnus incana*, *Alnus glutinosa*, *Betula pendula*

## Abstract

*Inonotus obliquus* grows in the Northern Hemisphere on some living broadleaved tree species as a pathogen, causing stem rot. In Estonia, the fungus is well known in the *Betula* species but can also be found on *Alnus*. Sterile conks of *I. obliquus* contain different bioactive compounds, but the quantitative and comparative research of these compounds in conks on different host species is limited. In the current work, *I. obliquus* was isolated and, evidently, determined from *Alnus incana* (L.) Moench., *Alnus glutinosa* (L.) Gaertn., and *Betula pendula* Roth, and the content of bioactive compounds in conks on these hosts were analysed. All the analysed conks sampled from *A. incana* and *B. pendula* contained betulin that varied from 111 to 159 µg/g. A significantly (*p* < 0.05) higher betulinic acid content was found in conks sampled from *A. incana* when compared with *B. pendula*: 474–635 and 20–132 µg/g, respectively. However, the conks from *Betula* were richer in total polyphenols, flavonols, and glucans. The content of inotodiol was quite similar in the conks from *A. incana* (7455–8961 µg/g) and *B. pendula* (7881–9057 µg/g). Also, no significant differences in the lanosterol content were found between the samples from these two tree species. To the best of our knowledge, this study is the first investigation of the chemical composition of *I. obliquus* parasitizing on *Alnus*. The results demonstrate that the bioactive compounds are promising in conks of *I. obliquus* growing not only on *Betula* but also on the *Alnus* species. It supports the opportunity to cultivate *I. obliquus,* also on the *Alnus* species, thus increasing the economic value of growing this tree species in forestry.

## 1. Introduction

Interest in non-timber forest products is increasing among landowners and foresters because consumer attention to bio- and other nature-based products has clearly risen, demonstrating, also, a growing trend in the national and international markets [1]. Naturally growing fungi are a good example of these non-timber forest products, and, already today, the cultivation of *I. obliquus* is evidently a higher interest of forest owners. The cultivation of fungi is one possibility, especially for small landowners, to increase their income from their land and forest.

Fungi belong to the most diverse organisms on earth [2], some of which have played an important role in human society since prehistoric times. The picking of edible mushrooms has been an important source of income in different areas of Europe [1]. Step by step, until recent years, they have become a strategic asset, not only in the politics of the conservation and management of ecosystems but have also retained their value as a resource for rural areas, for example, as in southern Europe [3].

Fungi with pharmacological potential represent a special interest. The history of picking one of them from nature in northern Asia and north-eastern Europe and using it for medical purposes reaches hundreds of years back. It is Chaga (*Inonotus obliquus* (Fr.) Pilát). By today, already the cultivation of *I. obliquus* has been recommended for integration into forestry, e.g., into the management of birch (*Betula* spp.) stands in Finland [4]. It is considered to become one way of alternative usage of forests instead of producing timber, especially in lower productive sites.

*Inonotus obliquus*, a basidiomycete, is naturally spread in the Northern Hemisphere [5]. It is a pathogenic fungus that grows mostly on *Betula* spp. but as well as on *Alnus* spp., which is more rear on some other broadleaved trees [6,7]. It is thought to be a real parasite, while it kills the host for fulfilling its entire lifecycle [8]. It means that the fungus forms its fruit bodies practically only on dead host trunks. However, the fungus also produces sterile conks (it means that these represent not fruitbodies of the fungus, but non-reproductive conks). The fungus is long known as Chaga to the North Asian indigenous peoples. The name “Chaga” actually originates from the language of a small Finno-Ugric nation, Komi, in North-Easternmost Europe but was disseminated to the world through the Russian language.

The conks of *I. obliquus* have traditionally been used in China, Korea, Japan, Russia, and the Baltics as an anti-tumor, antioxidant, anti-inflammatory, antibacterial, and hepatoprotective natural remedy. It promotes lipid metabolism and modulates cardiac function. In vitro, the properties of the conks were found to be mostly antioxidative, antiviral, and anti-inflammatory [9,10]. *Inonotus obliquus* can be even one of the potential candidates against the SARS-CoV-2 virus, but also several other antiviral effects of the conks have been shown in vitro and in vivo [11]. The triterpene and steroid compounds of these conks have demonstrated hepatoprotective, hypoglycemic, antifungal, antioxidant, and anti-inflammatory properties; in addition, they can regulate cholesterol biosynthesis [12]. In summary, *I. obliquus* is characterized as a promising natural material of biologically active substances against cancer and for immunotherapy [9,12,13,14,15,16,17,18].

In Estonian ethnomedicine, the conks of *I. obliquus* have been a popular folk anti-cancer remedy [19]. As was shown in the in vitro studies, the methanol extract of *I. obliquus* conks growing in Estonia exhibited the strongest cytotoxic effects against promyelocytic leukaemia and lung adenocarcinoma cells (the IC_50_ values were 32.2 and 38.0 μg/mL, respectively), and modest cytotoxic effects (41.3–57.7 μg/mL) against the cells of colon adenocarcinoma, liver hepatocellular carcinoma, oral epidermoid carcinoma, and prostate cancer cells [16].

Some recent studies concentrated on the resource incidence and ecology of *I. obliquus* on birch trees in North America [20] and on the inoculation and cultivation aspects of this fungus on birches *(Betula* spp.) in Finland [4]. However, little is known about the incidence and potential resources of the fungus on *Alnus incana* and *A. glutinosa*. These tree species are widely spread in northern Europe and are commonly used as fire and timber wood. *Alnus incana* has generally been considered to have a lower economic value and is suitable mostly for firewood. For example, in Estonia (with a forest land area of 2.33 mil. ha), *A. incana* dominated the stands cover by approximately 9% and *A. glutinosa* by 4% of the total forest area. At the same time, of the private forest area, *A. incana* covers 14.1% and *A. glutinosa* 4.3% [21].

As already mentioned, the conks of *I. obliquus* can also be found on *Alnus* spp. [5]. It is now worth considering that the collation data of fungi in Estonia revealed that the conks of *I. obliquus* are found on both alder species (*A. incana* and *A. glutinosa*), more frequently from *A. incana* [7]. At once, the question was raised about the value of the conks grown on alders in comparison to that on birches, which should estimate the cultivation potential of *I. obliquus* in alder stands.

Answering these questions was posed as the specific aim of this paper and solved by determining the bioactive compounds from the conks of *I. obliquus* grown on *Alnus incana*, *A. glutinosa*, and *B. pendula*. Additionally, as existing published articles about the conks of *I. obliquus* generally show a lack of quantitative data, this was also considered an essential motivation of the current study.

## 2. Material and Methods

### 2.1. Sample Sites and Fungal Isolation and Detection

Conks of *I. obliquus* were collected from Estonia and Finland between 21.12.2019 and 03.02.2020 (Table 1). The fungus was isolated directly from fresh conks to 2% malt-extract agar (Biolife, Milano, Italy) plates and sub-cultured several times to purify the isolates. Until biochemical analysis, the conks were kept in a freezer at –20 °C. The isolates were stored at −80 °C at the laboratory of forest pathology, Estonian University of Life Sciences.

The DNA of *I. obliquus* isolates was extracted using a GeneJET Genomic DNA Purification Kit (Thermo Scientific, Vilnius, Lithuania). The fungus was detected using ITS PCR primers ITS1-F [23] and ITS4 [24], carried out as described by Drenkhan et al. [25]. The PCR products were visualized on 1% agarose (SeaKem^®®^ LE Agarose, Lonza, Rockland, ME, USA) gels under UV light using a Quantum ST4-system (VilberLourmat SAS, Marne-la-all’ee, France). ITS-PCR products from the isolates were sequenced at the Estonian Biocentre in Tartu using the primer ITS5 [24]. The ITS sequences were edited using the BioEdit program, Version 7.2.5 [26] and deposited in a Genbank (see Table 1). BLAST searches for the fungal taxa confirmation were performed in the GenBank database (NCBI). All the pure cultures were deposited to the Fungal Culture Collection (TFC); the data are available in the data management and publishing platform, PlutoF (https://plutof.ut.ee/) (accessed on 20 July 2022).

### 2.2. Preparation and Bioactive Compound’s Detection in Inonotus Obliquus Conks

The collected conks were dried in a laboratory oven with a forced circulation Venticell (MMM Medcenter Einrichtungen GmbH, Planegg, Germany) at 50 °C, and all the material was ground to a coarse powder using a cutting mill Retch SM 300 with a 1 mm sieve (Retsch, Haan, Germany). The ground material was sealed airtight and stored in the freezer at −40 °C. For further analysis, a representative sample of 50 g was taken from the coarsely ground material and then further ground into a fine powder (particle size ~ 5 µm) using a Retsch MM400 homogenizer (Retsch, Haan, Germany) for 90 s at 30 Hz. Dry matter content in the samples was determined at 105 °C using a moisture analyser Precisa EM 120 HR (Precisa Gravimetrics AG, Dietikon, Switzerland).

For the determination of the α- and β-glucans, a Megazyme β-Glucan (Yeast & Mushroom) Assay Kit (Megazyme, Ireland) was used [27]. Then, 90 mg of the finely powdered material was dissolved in 2.0 mL of ice-cold 12 M sulphuric acid; after 2 h, 10 mL of water was added and suspension-mixed before being placed in a water bath at 99 °C and hydrolysed. Glucan fragments were quantitatively hydrolysed to glucose using specific enzymes (exo-1,3-β-glucanase and β-glucosidase, Megazyme, Wicklow, Ireland). The content of the β-glucans was determined by the glucose content produced by the splitting of the glucans, using only β-glycaemic enzymes and the GOPOD reagent (Megazyme, Wicklow, Ireland). The samples were analysed in two repetitions. All results are expressed as a % of the dry weight.

For the determination of free radical scavenging activity, the polyphenols and triterpenes content in the samples, 10 mL of 80% ethanol was added to 1 g of the finely ground sample and shaken on a Mini-rotator Multi RS-24 (Biosan, Riga, Latvia) for 24 h at room temperature. Then, 80% ethanol was selected for the extraction based on a pre-trial with different solvents (ethyl acetate, hexane, 80% EtOH, and a mixture of dichloromethane, MetOH, waterin propotion 5:3:2) (data not shown). Samples were centrifuged to remove insoluble material before analysis. Extracts were prepared in triplicate.

Measurements of the free radical scavenging activity were performed in duplicate using a 2.2-diphenyl-picrylhydrazyl (DPPH) assay with slight modifications [28]. Briefly, the gallic acid calibration for the analysis was prepared as follows: 0.125, 0.100, 0.0625, 0.050, 0.025, and 0.010 mg/mL. For the measurement, 0.1 mL of each standard was pipetted into a 4 mL spectrophotometer cuvette, to which 3.7 mL of the DPPH radical (103.9 µM) was added. The samples were incubated for 60 min in the dark at room temperature. The absorbance values of the samples were measured at 515 nm using a spectrophotometer (UV-1800, Shimadzu, Japan). The results were expressed in milligrams of gallic acid equivalent per gram of dry weight (mg GA eq./g).

The levels of the superoxide dismutase (SOD)-like free radical scavenging activity in the ethanol extracts were measured using a SOD Assay Kit-WST, according to the technical manual provided by Dojindo Molecular Technologies, Inc (Rockville, MD, USA). This assay relies on WST-1 (2-(4-iodophenyl)-3-(4-nitrophenyl-5-(2,4-disulfophenyl)-2H-tetrazolium, monosodium salt), which produces a water-soluble formazan dye upon reduction with superoxide ions, a reaction inhibited by the SOD. Briefly, in a 96-well microplate 20 μL, the sample was mixed with 200 μL of a WST working solution, and 20 μL of an enzyme working solution was added to each sample well. The plate was incubated at 37 °C for 20 min, and the OD was determined at 450 nm using a microplate reader, FLUOstar Omega (BMG Labtech, Ortenberg, Germany). One unit of the SOD activity was defined as the amount of enzyme having a 50% inhibitory effect on WST-1.

Qualitative and quantitative analyses were performed on a Shimadzu Nexera X2 UHPLC with a mass spectrometer, LCMS 8040 (Shimadzu Scientific Instruments, Kyoto, Japan). The UHPLC system was equipped with a binary solvent delivery pump, LC-30AD, an autosampler, Sil-30AC, a column oven, CTO-20AC, and a diode array detector, SPD-M20A. A reverse phase column, ACE Excel 3 (C18, PFP, 100 × 2.1 mm from ACE^®®^ Advanced Chromatography Technologies Ltd., Aberdeen, Scotland), and pre-column (Security Guard ULTRA, C18 from Phenomenex, Torrance, CA, USA) were used at 40 °C for the separation of the individual compounds. The flow rate of the mobile phase was 0.25 mL/min, and the injected sample size was 1 µL.

The mobile phases consisted of 1% formic acid in Milli-Q water (mobile phase A) and 1% formic acid in methanol (mobile phase B). The separation was carried out for 40 min under the following conditions: gradient 0–27 min, 15–80% B; 27–29 min, 80–95% B; 29–38 min, isocratic 95% B, and re-equilibration of the system with 15% B 6 min prior to the next injection. All samples were kept at 4 °C during the analysis.

The total polyphenol content, expressed as micrograms of chlorogenic acid equivalents per gram of dry weight (µg ChlA eq./g), and the total flavonols, expressed as micrograms of quercetin equivalents per gram of dry weight (µg Q eq./g), were quantified at the wavelengths of 280 nm and 360 nm, respectively.

Four triterpenoids (betulinic acid, betulin, lanosterol, and inotodiol) were identified by comparing the retention times, UV spectra, and parent and daughter ion masses with those of the standard compounds. MS data acquisitions were performed on LCMS 8040 with the ESI source operating in positive mode. All samples were analysed in triplicate, and the results were expressed as micrograms per gram of dry weight (µg/g).

The standards betulin, betulinic acid, and lanosterol were purchased from Cayman Chemical Company (Ann Arbor, MI, USA) and inotodiol from Aobious (Gloucester, MA USA). All other standards (chlorogenic acid and quercetin) and chemicals (formic acid and methanol) used were of an analytical grade and purchased from Sigma (Darmstadt Germany).

The statistical analysis was performed using a one-way ANOVA. Data analysis and visualization were aided by Daniel’s XL Toolbox adding for Excel, version 7.2.6, by Daniel Kraus, Würzburg, Germany [29].

## 3. Results and Discussion

As said, the conks of *I. obliquus* are well known in the living *Betula* species, but sometimes they are also formed on the *Alnus* species [5,6,7,30]. These conks contain different groups of bioactive substances [31], some of which have been studied quite a lot phytochemically and biologically, but the comparative quantitative research of these substances in conks formed on different host species is limited. This is the first appropriate comparative analysis of bioactive compounds in the conks of *I. obliquus* from *Betula pendula*, *Alnus incana*, and *A. glutinosa*.

### 3.1. Fungal Species Detection

*Inonotus obliquus* was isolated from the conks, and the fungus was evidently determined by molecular methods from three different host species: *Alnus incana*, *A. glutinosa,* and *Betula pendula* (see Table 1). Two fungal strains were isolated from *Alnus incana,* one from *A. glutinosa*, and three from *Betula pendula* (see Table 1). In the share of six isolates, five of them originated from Estonia and one from northern Finland. This study is the first where all chemically analysed conks strains (isolates) of *I. obliquus* were also deposited in a culture collection and GenBank for future research.

### 3.2. Determination of Bioactive Compounds

#### 3.2.1. Contents of Betulin and Betulinic Acid

Betulin and betulinic acid were present in all the samples, except for the conks collected from northern Finland (Betula FIN), where the named compounds were not identified. Although betulin was detected in the samples of *A. glutinosa* and Betula EST I (from *B. pendula*), the amount of betulin there was under the quantitation limit (Table 2). The content of betulin varied between 111 to 159 µg/g and was present in all the analysed conk samples of *Alnus incana* but varying from not quantifiable amounts to as high as 159 µg/g in the conk sample of *B. pendula* (Betula EST II).

The contents of the betulinic acid were statistically significantly (*p* < 0.05) different in all the analysed samples and varied from 20 µg/g in the strain Betula EST II to 635 µg/g in strain A. incana II (Table 2). The highest value of the content of betulinic acid was established in both of the conk samples of *A. incana* compared to the samples of the Estonian birch conks (474–635 and 20–132 µg/g, respectively). Relatively low was the content of betulinic acid in the sample of *A. glutinosa* but yet, more than two times higher than in the conk sample of Betula EST II (Table 2).

Betulin and betulinic acid are obtained through the oxidation of pentacyclic triterpene, which has been shown to have antiproliferative, anti-inflammatory, and antineovascular activity. It has an effect on various types of cancer, in vitro and in vivo [32]. Betulin, betulinic acid, lupenone, and other pentacyclic triterpenes have been found to be cytotoxic in vitro against the human adenocarcinoma and melanoma models [33]. Betulin, with low toxicity, shows activity against colon cancer cells [17]. Unfortunately, betulin has low aqueous solubility and variable bioavailability. Therefore, the nanosuspensions (sub-micron colloidal dispersions) of betulin were studied by John et al. [34], and, in this study, betulin demonstrated significantly higher cytotoxicity compared to betulinic acid (IC_50_ 38 and 70 µg/mL, respectively) against the resistant triple-negative human breast cancer cell line, MDA-MB-231.

#### 3.2.2. Contents of Inotodiol and Lanosterol

Inotodiol and lanosterol were present in all the samples and all three tested tree species. The inotodiol content varied from 6300 to 15,223 µg/g, being the lowest in the conks of *A. glutinosa* and the highest in the conks of the *B. pendula* sample from Finland. There were significant (*p* < 0.05) differences in the inotodiol content within the species among the samples collected from Estonia, but the inotodiol content had a quite similar range between the conk samples of *A. incana* (7455 to 8961 µg/g) and *B. pendula* (7881 to 9057 µg/g). the lanosterol content varied from 1021 to 2220 µg/g, and it was significantly higher (*p* < 0.05) in the originating from Finland conk of *B. pendula* compared to the samples from Estonia. No significant differences in the lanosterol content were established between the conk samples of Estonian origin from both the *Alnus* species and from *B. pendula*.

It is important to note that the use of various organic solvents may end with different results. For example, the conks originating from Canadian, Ukrainian, and French formulations yielded betulin, betulinic acid, and inotodiol of up to 1100, 53, and 465 mg/L, respectively [35]. In our study (Table 2), in contrast, betulin was in the lowest and inotodiol in the highest concentration. Other authors [36], who recently studied the biologically active substances isolated from different conks of *I. obliquus* by different extraction methods, found 0.6–17 mg% of betulin, 1.3–179 mg% of lanosterol, and 1.3–76 mg% of inotodiol.

Although the number of analysed samples in our investigation was relatively small, we can presume, based on these preliminary results, that inotodiol and lanosterol in north-eastern Europe can be found in the conks from all three of the investigated tree species. We also found the average positive correlation (r = 0.73) between the inotodiol and lanosterol contents in the analysed samples, which is logical due to the adjacent structures of these both tetracyclic triterpenoids. Although there was significant variation in the content of inotodiol and lanosterol compounds within different samples from the same tree species, the content of these compounds did not differ significantly between *A. incana* and *B. pendula*, indicating that both these tree species could be the real source of these compounds. Both of the triterpenes are important biologically active compounds, which obviously account for the manifestation of the *I. obliquus* conks’ antitumor effects through different pharmacological mechanisms [13,14]. Also, lanosterol and both of these terpenoids isolated from the conks on birch showed immunological activity [18,37].

#### 3.2.3. Contents of Total Polyphenols and Flavonols

In the current work, the conk samples growing on *B. pendula* contained about half more of the total polyphenols than the strains on *Alnus* spp. (663–841 and 277–477 µg/g, respectively; Table 2). In terms of total flavonols, birch conks are 5–8 times richer than *Alnus* spp. conks. However, the total content of polyphenols and flavonols does not necessarily correlate with the effect of the extract; the amount of individual key compounds in the natural material is more important.

Zhang et al. [38] concluded that there was a direct relationship between the phenolic compound concentration and antioxidant activity in the ethanol extracts, but polysaccharides played an important role in the antioxidant activities of the water extracts. Also, among the samples of *I. obliquus* conks studied by them, the ethanol extracts demonstrated the highest antioxidant properties and the best antioxidant activities with the highest total phenol and flavonoid content [38]. However, in the current study, the ethanol extracts of the *Alnus* spp. samples contained fewer polyphenols and showed weaker antioxidant activity than the *B. pendula* samples (Table 2).

#### 3.2.4. Contents of Glucans

Glucans are structural polysaccharides; beta-glucan plays a key role in the enhancement of immune function and the reduction of blood cholesterol and blood glucose levels [39]. The alpha-glucans’ content varied between 0.26% in the sample of Betula EST I to 0.60% in the sample of *A. glutinosa* (Table 3). No significant differences in the α-glucan content were between the analysed conks’ samples from *A. incana* and *B. pendula*, but the content of α-glucan in the sample from *A. glutinosa* was significantly different (*p* < 0.05). The β-glucans’ content varied between 2.34% in the sample from A. incana I to 6.75% in the sample from Betula EST I (Table 3). A significantly (*p* < 0.05) lower β-glucan content was observed in the conk samples from *Alnus* spp. compared to the samples from *Betula.* For example, in a previous study [39], the content of β-glucans in birch conks was 8.10–8.30%, depending on the extraction method.

The results showed (Table 2 and Table 3) that conks from birch contained more polyphenols, flavonols, and glucans than the samples grown on *Alnus* spp., but *A. incana*’s conks also contained inotodiol and lanosterol and even more betulinic acid than well-known birch conks.

The antioxidant activity of the ethanol extracts from the conks was determined by evaluating their DPPH free radical scavenging activity and was expressed as gallic acid equivalents per 100 g of dry material (mg GA eq./100 g). There were statistically significant differences between the samples from the different host species and within the samples from *B. pendula,* but no significant differences were observed within the samples from *A. incana.* Additionally, the samples from *A. incana* expressed the lowest radical scavenging activity within all of the analysed samples (Figure 1).

Glamočlija et al. [9] studied the antioxidant potential of aqueous and ethanolic extracts and showed that the DPPH scavenging activity of the aqueous extract of *I. obliquus* conks from Russia and Finland was double weaker, and the activity of the ethanolic extract from Finnish origin was about 40 times weaker than the extracts from conks growing in Thailand. In another study [40], the scavenging activity of the water, ethanol, and methanol extracts of *I. obliquus* conks on the DPPH radical was found to be concentration-dependent (12.50–200 mg/mL), respectively. The IC_50_ values for the water, ethanol and methanol extracts were 18.96, 16.25, and 24.90 mg/mL, respectively. Thus, in further studies, it is important to compare the effects of extracts made with different solvents.

The results from the SOD inhibitory assay were expressed as SOD units per gram of dry material (SOD U/g). The ethanol extracts from all the *Betula* samples expressed higher SOD activity than the rest of the samples (Figure 2), similarly to the DPPH assay. There was a strong positive correlation (r = 0.983) between the DPPH and SOD results.

Which of these substances would be more important in terms of the effects of bioactive compounds? In view of the biological activity of *I. obliquus* conks, especially their anti-cancer and immune-mediating effects on alder-grown conks, they may be a substitute for birch conks. The *Alnus* species are growing well in the northern Baltics. It would be one opportunity to increase the still comparatively low economic value of *A. incana* by cultivating this host *I. obliquus*. However, the cultivating profitability and production ability of conks on the *Alnus* species, but also the risk of the spread of this pathogen to other forest stands, needs deeper research in the future. Additionally, before a campaign for the cultivation of *I. obliquus* on the alder species, the biological activity of birch and alder conks should also be compared on the level of the effects of different cell lines of human cancers. A comparison of the chemical composition and biological activity of the two alder species (*A. incana* and *A. glutinosa*) on the base of many more individual conks from different regions of the Northern Hemisphere also requires further investigation.

## 4. Conclusions

The results contain the first comparative analysis of the content of several bioactive compounds in the so-called sterile conks of *I. obliquus* originating from *Alnus incana, A. glutinosa*, and *Betula pendula* in Estonia and Finland. In the current work, betulin, betulinic acid, lanosterol, inotodiol, α- and β-glucan, and the total polyphenols and flavonols in the conks of *I. obliquus* were analysed. The content of the betulinic acid had the highest value in the two conk samples of *A. incana* compared to the samples of the birch conks (474–635 and 20–132 µg/g, respectively). There was a significant variation in the content of inotodiol and lanosterol within the samples from the same tree species, but the content of these compounds did not differ significantly between *A. incana and B. pendula*. Betulin, α- and β-glucan, and the total polyphenols and flavonols were present in the conks of *Betula* and *Alnus*. The results indicate that *I. obliquus* conks, on both the *Betula* and *Alnus* tree species, could be the real source of these compounds, and it would be an opportunity also to cultivate *I. obliquus* on the *Alnus* species. However, further research is needed to establish the comparative effectiveness of fungal cultivation and profitability on different host tree species, to analyse the risk of the uncontrolled spread of the pathogen in forests, and to establish the effects of the biological activity of these substances on different cancer cell lines in medicine.

## Figures and Tables

**Figure 1 biomolecules-12-01178-f001:**
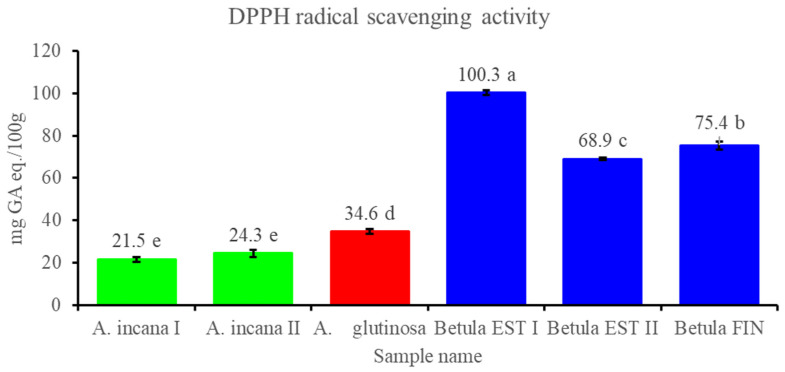
DPPH radical scavenging activity of 80% ethanol extracts from the conks expressed as gallic acid equivalents per 100 g of dry material (mg GA eq./100 g). All values are means, *n* = 3; mean values within a column with different letters are significantly different at *p* < 0.05.

**Figure 2 biomolecules-12-01178-f002:**
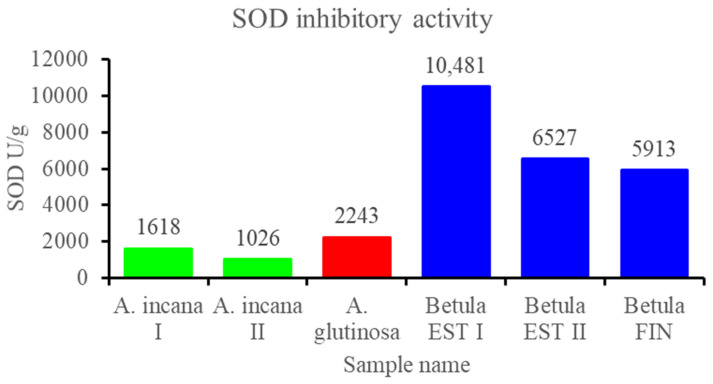
SOD inhibitory activity of 80% ethanol extracts from the conks expressed as SOD units per g of dry material (SOD U/g). One unit of SOD activity was defined as the amount of enzyme having a 50% inhibitory effect on WST-1.

**Table 1 biomolecules-12-01178-t001:** Origin and hosts of *Inonotus obliquus*, isolated from the conks and used in this study’s fungal strains.

Strain No ^a^	Host	Sample Name	Site Type ^b^	Sampling Site and Time	Geographical Coordinates	Fungal Culture Collection Code ^c^, GenBank Accession No ^d^
PAT29023	*Alnus incana*	A. incana I	*Aegopodium*	Estonia, Leemeti; 30.11.2019	N58.344944, E25.387528	TFC101255, OP019322
PAT29031	*A. incana*	A. incana II	*Aegopodium*	Estonia, Pedajamäe; 30.12.2019	N58.068278, E26.428222	TFC101256, OP019323
PAT29036	*A. glutinosa*	A. glutinosa	*Dryopteris*	Estonia, Jalametsa; 03.02.2020	N58.794250, E25.834667	TFC101257, OP019324
PAT29045	*Betula* *pendula*	Betula EST I	*Aegopodium*	Estonia, Kaavere; 21.12.2019	N58.903734, E26.446721	TFC101258, OP019325
PAT29043	*B. pendula*	Betula EST II	*Oxalis*	Estonia, Vissi; 14.01.2020	N58.160778, E26.732750	TFC101259, OP019326
PAT29027	*B. pendula*	Betula FIN	Unknown	Finland, Luosto; Jan. 2020	N67.115336, E26.899869	TFC101260, OP019327

^a^ In the collection of the Laboratory of Forest Pathology and Genetics of the Estonian University of Life Sciences; ^b^ according to Lõhmus [22]; ^c^ Tartu Fungal Collection in the Estonian University of Life Sciences, Estonia (TFC); ^d^ Genbank: https://www.ncbi.nlm.nih.gov/genbank/.

**Table 2 biomolecules-12-01178-t002:** Triterpenoids and polyphenolic compounds in the conk samples.

Sample Name	Betulin	Betulinic Acid	Inotodiol	Lanosterol	Total Polyphenols	Total Flavonols
	µg/g	µg/g	µg/g	µg/g	µg GA eq/g	µg Q eq/g
A. incana I	154 ± 8.6 ^a^	474 ± 10.8 ^b^	8961 ± 217.8 ^b^	1023 ± 29.0 ^c^	287 ± 11.1 ^d^	336 ± 22.9 ^c^
A. incana II	111 ± 6 ^b^	635 ± 34.7 ^a^	7455 ± 172.9 ^c^	1162 ± 21.1 ^b^	277 ± 7.4 ^d^	219 ± 9.2 ^d^
A. glutinosa	nq	49 ± 3.8 ^d^	6300 ± 90.1 ^d^	1248 ± 26.3 ^b^	477 ± 5.5 ^c^	364 ± 13.5 ^c^
Betula EST I	nq	132 ± 19.9 ^c^	9057 ± 343.5 ^b^	1080 ± 43.9 ^bc^	708 ± 7.5 ^b^	2879 ± 38.5 ^a^
Betula EST II	159 ± 26.3 ^a^	20 ± 5.0 ^e^	7881 ± 286.7 ^c^	1021 ± 24.0 ^c^	841 ± 18.7 ^a^	1137 ± 8.9 ^b^
Betula FIN	0	0	15223 ± 103.1 ^a^	2220 ± 29.8 ^a^	663 ± 24.4 ^b^	1126 ± 21.5 ^b^

nq—not quantified (signal to noise ratio < 10); µg GA eq./g: µg gallic acid equivalent/g; µg Q eq./g: µg quercetin equivalent/g. All values are means ± standard deviation, *n* = 3; mean values within a column with different letters (^a–e^) are significantly different at *p* < 0.05.

**Table 3 biomolecules-12-01178-t003:** Total content of glucan and α and β-glucan in the conk samples (in % per dry weight).

Sample Name	Total Glucan %	α-Glucan %	β-Glucan %
A. incana I	2.72 ± 0.64 ^d^	0.38 ± 0.03 ^e^	2.34 ± 0.66 ^c^
A. incana II	3.96 ± 0.11 ^c^	0.53 ± 0.01 ^b^	3.43 ± 0.09 ^c^
A. glutinosa	3.58 ± 0.08 ^d^	0.60 ± 0.01 ^a^	2.98 ± 0.09 ^c^
Betula EST I	7.01 ± 0.37 ^a^	0.26 ± 0.02 ^f^	6.75 ± 0.35 ^a^
Betula EST II	5.94 ± 0.3 ^b^	0.49 ± 0.01 ^c^	5.45 ± 0.29 ^b^
Betula FIN	6.04 ± 0.71 ^ab^	0.46 ± 0 ^d^	5.58 ± 0.71 ^ab^

All values are means ± standard deviation, *n* = 3; values within a column with different letters are significantly different at *p* < 0.05.

## Data Availability

Not applicable.

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
