# Peer review of "Comparative Analyses of Bioactive Compounds in Inonotus obliquus Conks Growing on Alnus and Betula"

_biomolecules, 2022, doi:10.3390/biom12091178_

Round 1

Reviewer 1 Report

The conks of Inonotus obliquus have traditionally been used in China, Korea, Japan, Russia,and the Baltics as an anti-tumor, antioxidant, anti-inflammatory, antibacterial, and hepatoprotective natural remedy. In general, the fungus parasitizes Betula species, and the bioactive compounds of the conks of I. obliquus grown on Betula species have been analyzed. Currently, the fungus can also be found on Alnus species in Estonia. However,  the bioactive compounds of the conks of I. obliquus grown on Alnus  species have never been determined. In present study, the bioactive compounds of I. obliquus conks growing on Alnus and  Betula were  compared. The results indicated that  the bioactive compounds from the conks of I. obliquus grown on Alnus  species are also promising. I think that this MS can be accepted for publication after minor revision. 

1. Line 42, I. Obliquus appears in the text for the first time and should be its full name. Thereafter, it should be abbreviated as I. Obliquus. Please check the full text.

2. Line 78, "Suming up" should changed to be "In sum,"

3. In Estonian ethnomedicine, 

4. Line 86, IC50 values were 32.2 and 38.0 μg/mL, respectively

5. Line 152, Please indicate the sources of exo-1,3-β-glucanase and β-glucosidase; 

Line 154 the sources of y β-glycaemic enzymes and the GOPOD reagent?

6. Line 216, In the share of six isolates, 

7. Line 361, In the current work,  8. In table 2, 8,.6?; The values in all table were suggested to  be expressed as "154 ±8.6a"

Author Response

Reviewer 1

Comments and Suggestions for Authors

The conks of Inonotus obliquus have traditionally been used in China, Korea, Japan, Russia, and the Baltics as an anti-tumor, antioxidant, anti-inflammatory, antibacterial, and hepatoprotective natural remedy. In general, the fungus parasitizes Betula species, and the bioactive compounds of the conks of I. obliquus grown on Betula species have been analyzed. Currently, the fungus can also be found on Alnus species in Estonia. However, the bioactive compounds of the conks of I. obliquus grown on Alnus  species have never been determined. In present study, the bioactive compounds of I. obliquus conks growing on Alnus and Betula were compared. The results indicated that the bioactive compounds from the conks of I. obliquus grown on Alnus species are also promising. I think that this MS can be accepted for publication after minor revision.

Response: thank you very much for this positive estimation to our MS.

  1. Line 42, I. Obliquus appears in the text for the first time and should be its full name. Thereafter, it should be abbreviated as I. Obliquus. Please check the full text.

Response: corrected as suggested, except in beginning of the sentence, because it should be in according to the English grammar.

  1. Line 78, "Suming up" should changed to be "In sum,"

Response: corrected as suggested.

  1. In Estonian ethnomedicine,

Response: corrected in our MS.

  1. Line 86, IC50 values were 32.2 and 38.0 μg/mL, respectively

Response: corrected in the MS.

  1. Line 152, Please indicate the sources of exo-1,3-β-glucanase and β-glucosidase;

Response: corrected, the source is added to the MS.

  1. Line 154 the sources of y β-glycaemic enzymes and the GOPOD reagent?

Response: corrected, source is added to the MS.

  1. Line 216, In the share of six isolates,

Response: corrected in the MS.

  1. Line 361, In the current work,

Response: corrected in the MS.

  1. In table 2, 8,.6?; The values in all table were suggested to be expressed as "154 ±8.6a"

Response: corrected.

Reviewer 2 Report

You have some information about the environmental condition in Finland that could explain the differences between sample collected from Estonia and Finland, especially for Betula?

The some differences between chemical composition of Betula trees could explain the differences between your Inonotus sample. Also, the soil composition could influences the results.

Try to evaluate the antioxidant activity by more specific tests.

Author Response

Reviewer 2

Comments and Suggestions for Authors

You have some information about the environmental condition in Finland that could explain the differences between sample collected from Estonia and Finland, especially for Betula?

Response: the single Finnish sample was collected in northern Finland and the distance from there to Estonia is more than 1000 km. Surely, there are environmental conditions different and it might affect also the formation of bioactive compounds in conks of I. obloquus, because the whole environment is different. But one sample does not allow yet to do any scientifically appropriate statements. Indeed, the environmental (incl. climatic, edaphic, etc.) impacts to the contents of bioactive compounds of I. obliquus conks are scientifically interesting and will be investigated in future, incl.  with a higher number of samples.    

The some differences between chemical composition of Betula trees could explain the differences between your Inonotus sample. Also, the soil composition could influences the results.

Response: thank you for these interesting notes and suggestions. The aim of the current work was not to analyse chemical compounds of host trees tissues. The purpose, in addition to the first glimpse into the chemistry of conks on Alnus spp. and comparisons of interesting us chemicals in conks on different hosts, was to demonstrate to the readers, what they may get when picking these conks in forest. Soil chemical compounds, as also very many different environmental characteristics,  may surely have some influence, but solving all of them would demand strongly higher number of samples.

Try to evaluate the antioxidant activity by more specific tests.

Response:  improved, superoxide dismutase (SOD) inhibitory activity data were added to the MS.
